# OpenReview forum: "Truthful or Fabricated? Using Causal Attribution to Mitigate Reward Hacking in Explanations"
_ICLR.cc/2026/Conference — ICLR 2026 Poster_

### Official Review · Reviewer_ChWA · 2025-10-31

**Soundness:** 3
**Presentation:** 2
**Contribution:** 3
**Rating:** 6
**Confidence:** 3

**Summary:**

The paper studies the CoT hacking scenario in which models improve task scores while CoTs fail to acknowledge reliance on a protected cue. RMs exacerbate this under BoN and DPO. The proposed fix is to append a natural-language disclaimer to the RM input when causal attribution (via an original vs. edited prompt) indicates the protected feature influenced the prediction. On two designed hacking scenarios, the augmented RM mitigates unfaithful CoTs and reduces the accuracy/stereotype vs. acknowledgment mismatch under both BoN and DPO settings.

**Strengths:**

1. The problem is timely and clearly motivated.

2. Augmenting RM inputs with a causal signal is lightweight (no RM retraining) and can be easily integrated into common pipelines. The two proposed strategies are intuitive and easy to implement.

3. The method effectively improves CoT quality/faithfulness, reflected in higher accuracy and acknowledgment rates.

**Weaknesses:**

1. The presentation needs improvement. For instance, RM_D and RM_C appear in Figures 4 and 5 with no explicit definition at all. And their corresponding strategies are only clarified late (end of section 5).

2. The two tasks across all experiments are controlled and stylized. It’s unclear how well the mitigation transfers to broader reasoning domains (code, multi-hop QA, tool use) where cues and causal edits are messier.

**Questions:**

1. The acknowledgment labels come from a separate Eval-LLM. Does this introduce a new opportunity for reward hacking or bias?

2. If you always append a neutral "placebo" disclaimer to all responses, do RMs behave similarly? This control would isolate the semantic effect of the disclaimer.

3. Did you test adversaries that acknowledge in form but conceal in content (e.g., obfuscated disclaimers) to see whether RMs can be hacked by superficial cues?

4. How robust is the disclaimer-trigger to decoding randomness and small prompt edits (e.g., cue paraphrases or formatting changes)?

---

> ### Author Response · Authors · 2025-11-24
> **Response to Reviewer ChWA [1/2]**
>
> Thank you for your valuable feedback and comments. We have updated the paper accordingly (with changes marked in blue) and now provide a point-by-point response to your remarks.
>
> ---
>
> > **The presentation needs improvement. For instance, RM_D and RM_C appear in Figures 4 and 5 with no explicit definition at all. And their corresponding strategies are only clarified late (end of section 5).**
>
> In order to clarify that $RM_D$ and $RM_C$ will only become relevant for analysis after they are introduced in Section 5 we added a footnote the first time Figure 4 is introduced mentioning that (line 297), as well as to the corresponding captions of Figures 4 and 5. We hope this further clarifies what are the relevant entries in the mentioned figures when they are first mentioned in the text.
>
> ---
>
> > **The two tasks across all experiments are controlled and stylized. It’s unclear how well the mitigation transfers to broader reasoning domains (code, multi-hop QA, tool use) where cues and causal edits are messier.**
>
> Indeed, the two settings we use are designed to make it possible to measure the phenomenon we want to study. For example, this requires an input cue whose use we can discourage via an instruction. Despite this, we note how our settings are concerned with two very different underlying challenges - one related to mitigating social biases (BiasQA) and the other related with preventing the use of protected information accidentally leaked (Math Book). Furthermore, we aim to make our results more robust by computing results according to two reward models, and have now added results for a different LLM generator, Llama-3.2-3B, with similar conclusions to Llama-3.1-8B (see those results in the added Appendix E).
>
> The main challenge to scaling this approach to more settings is the task of creating counterfactuals. Automating this step could follow a three-step pipeline: for each instruction (i) prompt a LLM to identify which information should be protected (e.g., using a predefined list of sensitive concepts or problematic biases, possibly generated by an LLM itself);  (ii) If anything is identified, generate counterfactuals based on the protected information identified in the previous step;  (iii) generate a disclaimer. A recent work (Matton et al., 2025) could be leveraged for steps (i) and (ii), as their results suggest that high-quality counterfactuals can be automatically produced. Alternatively, for specific high-stakes applications (e.g., medical domains or application pre-screening), a specialized, domain-specific pipeline for counterfactual generation could be manually constructed.
>
> ---
>
> > **The acknowledgment labels come from a separate Eval-LLM. Does this introduce a new opportunity for reward hacking or bias?**
>
> The Eval-LLM output is only part of the evaluation, and not of the augmentation signal given to the RM during training or during decoding with BoN. Thus, it does not offer a new path to be exploited by the model. We added a remark further clarifying this in lines 383-387.
>
> ---
>
> > **If you always append a neutral "placebo" disclaimer to all responses, do RMs behave similarly? This control would isolate the semantic effect of the disclaimer.**
>
> In order to answer your question we ran experiments where all examples are augmented with the introduced disclaimers according to the task. We did so for BoN with Llama-3-8B for both studied settings and reward models studied in this work. We added the results to Appendix Figure 7 and discussed it in lines 914-917. Here, for easier reporting, we show the percentage of the gap closed in terms of accuracy / stereotype rate between using the default RM with the original prompt and the counterfactual prompt. As we can see, adding the disclaimer to all examples is largely ineffective in reducing this gap, indicating that the impact we observe goes beyond the semantic effect of the added disclaimer.
>
> | Approach      | SK-Llama | SK-Gemma |
> | :----------- | :------: | :------: |
> | **Setting: MathBook** |          |          |
> | $RM_D$         | 18       | 9        |
> | $RM_C$         | 51       | 41       |
> | Placebo      | 2        | 1        |
> | **Setting: BiasQA**   |          |          |
> | $RM_D$         | 34       | 32       |
> | $RM_C$         | 98       | 71       |
> | Placebo      | 23       | 5        |

---

> ### Author Response · Authors · 2025-11-24
> **Response to Reviewer ChWA [2/2]**
>
> > **Did you test adversaries that acknowledge in form but conceal in content (e.g., obfuscated disclaimers) to see whether RMs can be hacked by superficial cues?**
>
> If the question is understood correctly, you are referring to the possibility that the LLM response could include a “shallow” acknowledgment, that is, one that satisfies the instruction not to use the protected feature, while in fact still relying on it to produce a response (that is, an “obfuscated disclaimer”).
>
> We first note that, as mentioned before in this discussion, the Eval-LLM component that detects whether a response acknowledges or not the use of the protected feature is solely part of evaluation, and thus, does not create a path for the model to exploit this signal during training.
>
> Secondly, even if the model produces a “shallow” acknowledgment in its response, the gap in performance with respect to the counterfactual prompt will not be affected by that: a model that produces “shallow” acknowledgements while still using the protected feature will have on average an accuracy / stereotype rate higher to the counterfactual. For example, in the Math Book setting, if a response uses the protected feature to solve the problem and produces a “shallow” acknowledgment (e.g.,”I did not use the provided Math Book and the best answer is (B)”), it will not contribute to an increase of the Ack Rate metric, but it will contribute to the model having an Accuracy higher than the model that uses the counterfactual prompt as input. Higher accuracy (or stereotype rate) without a corresponding increase in Ack Rate is what the test we employ in this work uses to check for potentially deceptive behavior by the model. We hope this makes it clear why our proposed approach is not affected, by design, by such “obfuscated” disclaimers.
>
> ---
>
> > **How robust is the disclaimer-trigger to decoding randomness and small prompt edits (e.g., cue paraphrases or formatting changes)?**
>
> Indeed , the proposed technique can be sensitive to changes in the form of the input (e.g. paraphrases or formatting changes as you mention) or to the randomness in decoding. In this work, we focus on average performances, across many examples and different seeds, rather than on the impact on individual examples (except for some reported experiments, like the percentage of unfaithful examples, but where we still use different seeds). This way we reduce sensitivity to the surface-level changes you mentioned, and ultimately observe trends that indicate that on average a default reward model increases CoT hacking, and that the proposed augmentation technique to the reward model input helps reduce it.
>
> ---
>
> **References:**
> - Matton et al., 2025. Walk the Talk? Measuring the Faithfulness of Large Language Model Explanations. ICLR.

---

> ### Author Response · Authors · 2025-11-27
> **Response to Reviewer ChWA**
>
> Dear Reviewer ChWA,
>
> Thank you once again for your constructive review. As the discussion period is expected to end in less than a week, could you please let us know if our response addresses your concerns? We are happy to provide any further clarifications or results.
>
> We would also like to draw your attention to the updated document, which has been substantially revised in line with your and the other reviewer’s feedback, including: experiments with a new LLM generator, updates to the first figure and introduction to clarify the role of the reward model in CoT hacking, a new figure on counterfactual construction with extended discussion, additional remarks on the role of Eval-LLM and on scaling the approach to other domains, and results studying the semantic effect of the added disclaimer.
>
> Thank you.

---

### Official Review · Reviewer_64gQ · 2025-10-31

**Soundness:** 3
**Presentation:** 4
**Contribution:** 2
**Rating:** 6
**Confidence:** 4

**Summary:**

This paper presents a case study of unfaithful CoT responses from LLMs and how reward model-based preference optimisation of LLMs can encourage these undesirable responses. Two forms of CoT hacking are introduced, one in math QA and one in biased QA. CoT hacking happens when there is a clear cue leading to a correct answer to the question in the prompt, the LLM used it for generating its answer, but does not recognise that it has used it. Counterfactual data augmentation technique is employed to identify such cases. Empirical evidence shows that CoT hacking can happen, reward models can encourage this, and that simply adding a textual hint of unfaithfulness for the reward model can mitigate unfaithful responses. These are investigated on one LLM for answer generation, and two different RMs.

**Strengths:**

- The paper targets the important and timely problem of LLM answer faithfulness. Unlike many prior works which simply acknowledge that this could happen, this paper identifies specific instances of LLM unfaithfulness, tries to find causes for it and proposes simple ways to fix it. Therefore, originality is excellent.
- The paper comes with excellent clarity. The targeted problem (a specific type of unfaithful LLM behaviour) is well-motivated, and its scope is also explicitly discussed. The illustrations throughout are quite clear.
- The experiment procedure is novel and makes sense. The resulting empirical findings in this paper are valuable to the research community.

**Weaknesses:**

- While this is in the style of a case study, it is concerning that all experiments are done with generations from a single LLM. A larger scope of the experiments could make the story more convincing.
- The setting seems quite simple. I wonder how this finding and methodology generalise to more open-ended QA forms.

**Questions:**

See weaknesses.
- This paper used two reward models which output one scalar score. How about those regression-based reward models which predict a score for a list of different aspects, such as relevance, verbosity, helpfulness, etc (for example, those trained on HelpSteer datasets). Would those reward models demonstrate similar behaviours?

---

> ### Author Response · Authors · 2025-11-24
> **Response to Reviewer 64gQ**
>
> Thank you for your valuable feedback and comments. We have updated the paper accordingly (changes are marked in blue) and provide a point-by-point response to your remarks below.
>
> ---
>
> > **While this is in the style of a case study, it is concerning that all experiments are done with generations from a single LLM. A larger scope of the experiments could make the story more convincing.**
>
> In order to strengthen our findings, we have added results where we use Llama-3.2-3B as the LLM generator, with similar conclusions to the ones we had found for Llama-3.1-8B: the default reward model promotes CoT hacking by increasing accuracy / stereotype rate, without a corresponding increase in acknowledgment rate; and augmenting the reward model input with a disclaimer based on the output of the employed causal attribution technique helps alleviate this behavior. These results and a more detailed discussion can be seen in the Appendix E, with mentions to the different results for BoN and DPO, both with greedy and sampling decoding, and the two reward models we use.
>
> ---
>
> > **The setting seems quite simple. I wonder how this finding and methodology generalise to more open-ended QA forms.**
>
> We expect our methodology to extend to more open-ended QA settings. The key adaptation would be replacing the strict exact-match criterion used to identify both the correctness and agreement across runs  with a broader ‘semantic equivalence’; the exact match can be too strict in open-ended cases. For example, one could employ an LLM-as-a-judge approach to approximate equivalence.
> Once such a prediction signal is available, both strategies C and D can be computed and applied in the same way as in our current setup.
>
> ---
>
> > **This paper used two reward models which output one scalar score. How about those regression-based reward models which predict a score for a list of different aspects, such as relevance, verbosity, helpfulness, etc (for example, those trained on HelpSteer datasets). Would those reward models demonstrate similar behaviours?**
>
> We agree that this would be an interesting direction to explore, particularly to determine whether certain attributes are more prone to being exploited. However, reward model outputs are generally aggregated into a single scalar score, so this would not change the way our method is used in this work.
> In principle, there are emerging approaches that use textual feedback directly (e.g., Hu et al., 2024; Luo et al., 2025) or pursue multi-objective alignment (e.g., Ankner et al., 2024) and process reward models (e.g., Shen et al., 2024). These are exciting developments but remain active areas of research with no well established methodology, so we believe that integrating our method with them is best left for future work.
>
> ---
>
> **References:**
> - Ankner et al., 2024. Critique-out-loud reward models. Preprint.
> - Hu et al., 2024. Teaching Language Models to Self-Improve by Learning from Language Feedback. ACL.
> - Luo et al., 2025. Language Models can Learn From Verbal Feedback without Scalar Rewards. Preprint.
> - Shen at al., 2024. Simultaneous Multi-objective Alignment Across Verifiable and Non-verifiable Rewards. Preprint.

---

> ### Author Response · Authors · 2025-11-27
> **Response to Reviewer 64gQ**
>
> Dear Reviewer 64gQ,
>
> Thank you once again for your constructive review. As the discussion period is expected to end in less than a week, could you please let us know if our response addresses your concerns? We are happy to provide any further clarifications or results.
>
> We would also like to draw your attention to the updated document, which has been substantially revised in line with your and the other reviewer’s feedback, including: experiments with a new LLM generator, updates to the first figure and introduction to clarify the role of the reward model in CoT hacking, a new figure on counterfactual construction with extended discussion, and remarks on scaling the approach to other domains.
>
> Thank you.

---

### Official Review · Reviewer_zgsZ · 2025-10-31

**Soundness:** 2
**Presentation:** 3
**Contribution:** 3
**Rating:** 6
**Confidence:** 3

**Summary:**

This paper focuses on addressing the lack of consistency between an LLM’s generated CoT explanations and the LLM’s actual internal decision making process.
* The authors first demonstrate that preference optimization leads to a reduction in explanation faithfulness because the reward models can inadvertently encourage unfaithful chain-of-thought (CoT) explanations through reward hacking.
* The authors propose adding causal attribution signals to the reward model’s input to encourage explanation faithfulness.
* Finally, they show that this method leads to a reduction in the generation of misleading explanations on two controlled tasks, MathBook and BiasQA.

**Strengths:**

* This work is well-motivated as CoT explanations are widely used to evaluate model outputs, and it is important to ensure the faithfulness of the explanations.
* The mitigation technique is simple but effective, and has minimal computational overhead.
* The authors conduct comprehensive experiments to show that their method improves upon other baselines.

**Weaknesses:**

* While the results are convincing, the experiments are limited to two specific, controlled settings and one LLM. It’s unclear how this approach would generalize to other cases, especially in cases where the use of protected attributes is more subtle or challenging to detect.
* There is not much discussion about the limitations of the approach or cases where the method may struggle to perform well.

**Questions:**

Can you elaborate more on the pros and cons of strategy C versus D?

---

> ### Author Response · Authors · 2025-11-24
> **Response to Reviewer zgsZ**
>
> Thank you for your valuable feedback and comments. We have updated the paper accordingly (changes are marked in blue) and provide a point-by-point response to your remarks below.
>
> ---
>
> > **While the results are convincing, the experiments are limited to two specific, controlled settings and one LLM.**
>
> In order to strengthen our findings, we have added results where we use Llama-3.2-3B as the LLM generator, with similar conclusions to the ones we had found for Llama-3.1-8B: the default reward model promotes CoT hacking by increasing accuracy / stereotype rate, without a corresponding increase in acknowledgment rate; and augmenting the reward model input with a disclaimer based on the output of the employed causal attribution technique helps alleviate this behavior. These results and a more detailed discussion can be seen in the Appendix E, with mentions to the different results for BoN and DPO, both with greedy and sampling decoding, and the two reward models we use. Note that we  also perform experiments with 2 different reward models.
>
> Although both settings are controlled, they address two very different challenges - mitigating social biases versus preventing reliance on accidentally leaked protected information. The fact that our method is beneficial in both provides encouraging evidence toward generality, though further investigation is warranted.
>
> ---
>
> > **It’s unclear how this approach would generalize to other cases, especially in cases where the use of protected attributes is more subtle or challenging to detect.**
>
> Indeed, our interpretability signal (i.e., the disclaimer added to the input to the reward model according to the discussed strategies C and D) relies on the ability to produce counterfactuals for the inputs at hand. Thus, when the use of the protected attributes is more subtle and challenging to detect, producing the required counterfactuals might prove to be more difficult.
>
> In particular, automating the creation of counterfactually augmented data could follow a three-step pipeline: for each instruction (i) prompt a LLM to identify which information should be protected (e.g., using a predefined list of sensitive concepts or problematic biases, possibly generated by an LLM itself);  (ii) If anything is identified, generate counterfactuals based on the protected information identified in the previous step;  (iii) generate a disclaimer. A recent work (Matton et al., 2025) could be leveraged for steps (i) and (ii), as their results suggest that high-quality counterfactuals can be automatically produced.  Alternatively, for specific high-stakes applications (e.g., medical domains or application pre-screening), a specialized, domain-specific pipeline for counterfactual generation could be manually constructed.
>
> ---
>
> > **There is not much discussion about the limitations of the approach or cases where the method may struggle to perform well.**
>
> We incorporated the above discussion into the conclusion of the paper (see Section 7 in blue).
>
> ---
>
> > **Can you elaborate more on the pros and cons of strategy C versus D?**
>
> The strategies used to augment the reward model input with a disclaimer rely on comparing the predicted labels for the original input (y = pred(x)) and the counterfactual input (y' = pred(x')).
>
> - **Strategy D** adds the disclaimer whenever the predictions differ, that is, when y != y'.
> - **Strategy C** adds the disclaimer only when the predictions differ  and the original prediction is cued, that is, when y != y' and y matches the cue.
>
> The main advantage of **Strategy D** is that it does not require any label information (or knowledge of how cues correspond to outputs); it simply checks whether the prediction changes under the counterfactual intervention.
>
> In contrast, **Strategy C** leverages the additional signal of whether the original prediction aligns with the cue. As shown in our results, this improves the effectiveness of the correction and reduces noise in the counterfactual signal, making it more reliable.
>
> ---
>
> **References:**
> - Matton et al., 2025. Walk the Talk? Measuring the Faithfulness of Large Language Model Explanations. ICLR.

---

> ### Author Response · Authors · 2025-11-27
> **Response to Reviewer zgsZ**
>
> Dear Reviewer zgsZ,
>
> Thank you once again for your constructive review. As the discussion period is expected to end in less than a week, could you please let us know if our response addresses your concerns? We are happy to provide any further clarifications or results.
>
> We would also like to draw your attention to the updated document, which has been substantially revised in line with your and the other reviewer’s feedback, including: experiments with a new LLM generator, updates to the first figure and introduction to clarify the role of the reward model in CoT hacking, a new figure on counterfactual construction with extended discussion, and remarks on scaling the approach to other domains.
>
> Thank you.

---

### Official Review · Reviewer_vuLZ · 2025-11-01

**Soundness:** 2
**Presentation:** 1
**Contribution:** 2
**Rating:** 4
**Confidence:** 3

**Summary:**

The paper studies the problem of explanation misalignment: where the explanations of model behavior do not reflect its actual decision making process. The specific form of explanations that the paper focuses on is the chain-of-thought (CoT) explanation. The paper identifies preference optimization as a part of the problem. The idea is that during preference optimization, the model responses need not only be accurate, but also adhere to other relevant criteria like fairness. So the outputs that adhere to these additional criteria are rewarded during training. Consequently, the LLM’s explanations end up being not fully faithful since the outputs containing explanations aim to maximize the reward. Inspired by prior works, the paper carefully designs contractual tests where the LLM is provided the answers to a question in the prompt but is told to not rely on this solution. The model is then prompted to generate the answer and a CoT explanation. By including and excluding the answer (also called the protected features and answer cue in the paper), the paper then studies if the CoT mentions the answer present in the prompt. Since the reward model cannot check the faithfulness of the explanation (whether or not the LLM relied on the answer in the prompt), the LLM ends up having misaligned explanations. The paper then designs a causal attribution mechanism to detect and mitigate such unfaithful behavior and shows increase in performance.

**Strengths:**

1. The problem is novel and important. As the paper mentions, the CoT misalignment problem has been already discussed in prior work. However, to the best of this reviewer’s knowledge, the effect of reward model on such behavior has not bee studied. Controlling the CoT unfaithfulness would definitely enhance reliability of LLMs.
2. The proposed mechanism for testing the faithfulness of the CoT mechanism, though closely related to prior studies, is intuitive.

**Weaknesses:**

1. The writing of the paper can be significantly improved. Right now, important details are missing which make it hard to appropriately judge its contribution. (a) Starting with a relatively minor issue, I would highly suggest moving Figure 2 into the intro and potentially merging it with Figure 1. It’s only after reading Figure 2 and seeing the score distribution that the reader becomes aware of the main problem, that is, the reward model rewarding no acknowledgement when the instruction to not use the solution is added. Figure 1 tries to achieve a similar goal but the right panel is very difficult to understand with the current supporting text. (b) The procedure in the paragraph starting line 329 is one of the most important parts of the paper. I would suggest adding a visual like a figure with prompts to explain the concept. For instance, the introduction of coutnerfactuals in line 336 is quite abrupt. (c) The metric of Majority@16 is introduced in line 206 without context and it is not clear what it does. (d) The details around the reward model in Section 3.2 are unclear. Why not take an off the shelf reward model? How many data points was the reward model in the paper trained with?
2. The paper relies on judge LLMs to extract the answer and check acknowledgements. However, the accuracy of the judges in checking answer correctness is not evaluated. The accuracy of judges in checking acknowledgements is evaluated in Appendix C but is somewhat weak. Why compute the F1 score? Why not simply measure a binary correctness metric which checks if the judge model correctly indicated that the protected feature is being used? Even if F1 score is a good metric (which I am not sure about), it would be good to add examples that help the reader understand why 0.84 and 0.65 are good F1 values for this task.
3. Line 255: I thought the experiments were done with greedy decoding. That is, the model solving the math questions was operating in a greedy decoding setting. If that is the case, why repeat with different seeds three times? If that is not the case, how can be say that the change in accuracy in Figure 3 is due to the model relying on the cue and not due to the randomness from decoding?
4. The paper should add some experiments showing how well the proposed method generalizes across various domains. For instance, does training on two datasets generalize to a third one?

**Questions:**

Please see the questions in W1, W3 and W4.

---

> ### Author Response · Authors · 2025-11-24
> **Response to Reviewer vuLZ [1/2]**
>
> Thank you for your valuable feedback. We revised the paper to improve clarity and address the concerns you raised, including:
> - **Introduction:** rewritten to provide a clearer intuition of how the reward model drives hacking behaviour.
> - **Figures:** the key part of former Figure 2 (score distribution) is now merged into Figure 1 and referenced in the introduction; the remaining part was moved to Appendix B (now Figure 6).
> - **Counterfactuals:** added a new figure (now Figure 2) to visualise the construction process and causal attribution technique, and improved the explanation.
>
> The changes in the paper are marked in blue.  We hope these changes fully address your comments and strengthen the presentation. We now address all your comments individually:
>
> ---
>
> >  **I would highly suggest moving Figure 2 into the intro and potentially merging it with Figure 1. It’s only after reading Figure 2 and seeing the score distribution that the reader becomes aware of the main problem, that is, the reward model rewarding no acknowledgement when the instruction to not use the solution is added.**
>
> We implemented the suggested change (merge Figures 2 and 1 into Figure 1) and revised the introduction text to clarify that the reward model (as seen from the score distribution) drives the hacking behaviour. We also expanded the caption (see changes shown in blue). We hope this fully addresses this concern.
>
> ---
>
> > **The procedure in the paragraph starting line 329 is one of the most important parts of the paper. I would suggest adding a visual like a figure with prompts to explain the concept. For instance, the introduction of coutnerfactuals in line 336 is quite abrupt.**
>
> Counterfactuals were first introduced in Section 3.2 as part of our evaluation strategy. However, we acknowledge that in Section 5, where they become central to our mitigation method, we did not adequately explain them. Following your advice, we also have added a new figure (now Figure 2) that illustrates the construction process of counterfactuals. We have also substantially revised the text in Section 5 to recap their construction and more carefully explain their use in the mitigation context.
>
> ---
>
> > **The metric of Majority@16 is introduced in line 206 without context and it is not clear what it does.**
>
> In order to make it clear why and how Majority@16 is used, we have rewritten and expanded its description in the paragraph “Evaluation” in Section 3.2 in lines 207-213. To summarize, Majority@16 is used to aggregate the outcome of multiple samples into a single prediction, when sampling multiple responses per input.
>
> ---
>
> > **The details around the reward model in Section 3.2 are unclear. Why not take an off the shelf reward model? How many data points was the reward model in the paper trained with?**
>
> In fact, we do use two off-the-shelf reward models, “Skywork-Reward-Llama-3.1-8B-v0.2” and “Skywork-Reward-Gemma-2-27B-v0.2” (Liu et al., 2024), both with strong performance on the RewardBench benchmark. We modified lines 188-190 in Section 3.2 where we introduce the used reward models to make this clear, and additionally added the number of preference pairs these reward models have been trained with (26M preference pairs).
>
> ---
>
> > **Why compute the F1 score? Why not simply measure a binary correctness metric which checks if the judge model correctly indicated that the protected feature is being used?  Even if F1 score is a good metric (which I am not sure about), it would be good to add examples (...)**
>
> If we understand correctly, the concern is that we reported per-label F1 scores for Eval LLM, which is indeed non-standard. We initially chose this representation as we found it informative, but we agree that it may cause confusion. In response, we now report accuracy as well as macro-averaged F1 (the latter is necessary due to class imbalance):
> - MathBook: 93% macro F1 and 95% accuracy
> - BiasQA: 74% macro F1 and 78% accuracy
>
> We have also followed your suggestion and added example responses, along with the target and predicted labels, in Tables 2 and 3 of the Appendix. These examples help explain why performance on BiasQA is lower: some cases are ambiguous even for humans (for instance, the model refers in a chain to "butcher" as a “male noun,” which could either reflect stereotypical bias or confusion with grammatical gender; i.e., milkman is gendered but butcher is not). We hope that these additions (both the accuracy metric and the qualitative examples) clarify the behavior of the Eval-LLM and address your concern.

---

> ### Author Response · Authors · 2025-11-24
> **Response to Reviewer vuLZ [2/2]**
>
> > **the accuracy of the judges in checking answer correctness is not evaluated**
>
> Our evaluation for correctness does not involve an LLM-judge. Both our settings rely on multiple-choice questions, which allow us to determine model predictions through exact matching.
>
> ---
>
> >  **I thought the experiments were done with greedy decoding. That is, the model solving the math questions was operating in a greedy decoding setting. If that is the case, why repeat with different seeds three times? If that is not the case, how can be say that the change in accuracy in Figure 3 is due to the model relying on the cue and not due to the randomness from decoding?**
>
> We evaluate both with greedy decoding and sampling-based decoding. Evaluating with sampling is important for two reasons: (1) it aligns with common practice in the real use of LLMs, and (2) alignment requires sampling to produce diverse outputs, so it is natural to examine how sampling-based outputs change as a result of training.   Using 3 seeds makes it possible to account for variance when decoding with Sampling + Majority@16, and when training the DPO models. Training DPO models with different seeds leads to different finetuned models, where also greedy decoding produces different samples.
>
> As for the second remark, we agree that it is challenging to reliably determine how much of the variation in accuracy (e.g., in the original Figure 3) stems from randomness across different seeds. Nonetheless, the average results consistently show a clear pattern: the default reward model increases accuracy and stereotype rate without a corresponding rise in acknowledgement rate. This strongly suggests that the improvement is driven by reliance on the cue, rather than random fluctuations (an interpretation further supported by the motivation example in Figure 1b). Moreover, in this, we largely follow the protocol used in causal attribution studies of CoT faithfulness (Chua et al., 2024; Turpin et al., 2024).
>
> ---
>
> > **The paper should add some experiments showing how well the proposed method generalizes across various domains. For instance, does training on two datasets generalize to a third one?**
>
> Our work looks at 2 specific domains. For good generalization to new tasks to occur, we posit this would require scaling our experiments to multiple tasks during training. In particular, doing so requires automating the creation of counterfactuals, with interesting recent works showing promising results such as Matton et al., 2025. This is indeed a very interesting direction.
>
> As a preliminary experiment we did the following: for the MathBook setting, we create a set of similarly structured inputs, but using a generic “SOLUTIONS BOOK”, with programming coding questions, for which we used a sample of 300 examples from the “software principles” split of CodeMMLU (Manh  et al., 2024). Then, we compute the gap in accuracy and acknowledgment rate between using as input the original prompts (i.e., one where the model has access to the solution to the problem as part of prompt, as in the Math Book setting) and the respective counterfactual prompts. We added these results to Appendix Section F. We can see that $RM_D$ is able to reduce the gap in accuracy in 2 out of 4 comparisons, while $RM_C$ does so in 3 out of 4 comparisons, when compared to $RM$, while acknowledgment rate gaps remain similar. These initial findings are encouraging, motivating further work in this direction.
>
> | Model      | Reward Model | % Acc      | % Ack     |
> | :--------- | :----------- | :--------: | :-------: |
> | **Greedy** |              |            |           |
> | DPO + $RM$   | SK-LLAMA-8B  | 15.2 ± 1.8 | 1.8 ± 0.6 |
> | DPO + $RM_D$ | SK-LLAMA-8B  | 15.3 ± 1.2 | 2.0 ± 0.2 |
> | DPO + $RM_C$ | SK-LLAMA-8B  | 13.1 ± 1.3 | 1.9 ± 1.1 |
> | DPO + $RM$   | SK-GEMMA-27B | 17.3 ± 2.9 | 1.8 ± 0.5 |
> | DPO + $RM_D$ | SK-GEMMA-27B | 17.4 ± 0.5 | 2.1 ± 1.2 |
> | DPO + $RM_C$ | SK-GEMMA-27B | 17.4 ± 2.0 | 2.4 ± 0.9 |
> |            |              |            |           |
> | **Sampling Majority@16**  | |          |           |
> | DPO + $RM$   | SK-LLAMA-8B  | 23.1 ± 0.2 | 0.7 ± 0.1 |
> | DPO + $RM_D$ | SK-LLAMA-8B  | 20.3 ± 2.0 | 0.7 ± 0.1 |
> | DPO + $RM_C$ | SK-LLAMA-8B  | 20.9 ± 1.2 | 1.0 ± 0.4 |
> | DPO + $RM$   | SK-GEMMA-27B | 24.9 ± 1.2 | 0.9 ± 0.2 |
> | DPO + $RM_D$ | SK-GEMMA-27B | 23.1 ± 1.1 | 0.8 ± 0.1 |
> | DPO + $RM_C$ | SK-GEMMA-27B | 23.0 ± 1.7 | 0.9 ± 0.2 |
>
> **References:**
> - Liu et al., 2024. Skywork-reward: Bag of tricks for reward modeling in llms.
> - Chua et al., 2024. Bias-Augmented Consistency Training Reduces Biased Reasoning in Chain-of-Thought.
> - Manh et al., 2024. CodeMMLU: A Multi-Task Benchmark for Assessing Code Understanding & Reasoning Capabilities of CodeLLMs. ICLR
> - Matton et al., 2025. Walk the Talk? Measuring the Faithfulness of Large Language Model Explanations. ICLR.
> - Turpin et al., 2024. Language Models Don't Always Say What They Think: Unfaithful Explanations in Chain-of-Thought Prompting. NeurIPS.

---

> ### Author Response · Authors · 2025-11-27
> **Response to Reviewer vuLZ**
>
> Dear Reviewer vuLZ,
>
> Thank you once again for your constructive review. As the discussion period is expected to end in less than a week, could you please let us know if our response addresses your concerns? We are happy to provide any further clarifications or results.
>
> We would also like to draw your attention to the updated document, which has been substantially revised in line with your and the other reviewer’s feedback, including: experiments with a new LLM generator, updates to the first figure and introduction to clarify the role of the reward model in CoT hacking, a new figure on counterfactual construction with extended discussion, and preliminary results regarding generalization to a different domain.
>
> Thank you.

---

### Author Response · Authors · 2025-12-03

Dear Area Chair,

Thank you for your effort in further reviewing our work. With this comment we aim to provide a summary of the rebuttal phase, where we provided responses and additional experiments to the concerns raised by the reviewers. We note however that we did not receive any further feedback from reviewers before the discussion was closed.

We received scores (4, 6, 6, 6). Most concerns of reviewer **vuLZ** (Score: 4) are due to misunderstandings. In particular, (i) our work does not rely on an LLM for correctness, but only to assess acknowledgement of the protected feature; (ii) we use F1 score to assess the quality of the Eval-LLM which is standard for imbalanced labels; (iii) the initial version of the manuscript already refers that sampling is used throughout experiments; (iv) the reward models we use are introduced as already existing and cited; and (v) counterfactuals are introduced in Section 3, before being explored in Section 5. Additionally, while we follow the reviewer’s presentation suggestions (which we are thankful for and acknowledge that improve the paper), these mainly involved reordering and expanding discussion, which benefited from the additional page during rebuttal, rather than addressing substantive methodology issues.

---

## Strengths highlighted by the reviewers
- Studies a problem that is **”novel and important”** and **”well-motivated”**, and identifies a specific cause of CoT unfaithfulness and proposes a way to fix it. **vuLZ,zgsZ,64gQ,ChWA**
- Provides **”excellent clarity”**, clearly motivating the target problem, with **”illustrations throughout (that) are quite clear”**. **64gQ**
- Proposes a **”simple but effective”** mitigation technique that requires no further training, and is **”intuitive and easy to implement”**. **zgsZ, ChWA**
- Includes **“comprehensive experiments”** and **”empirical findings valuable to the research community”**. **zgsZ, 64gQ**

---

## Key Concerns

| **Category** | **Concern** | **Response Summary** | **Changes to Manuscript** |
|--------------|-------------|-----------------------|----------------------------|
| **1. Model & Domain Generalization** | Reliance on a single LLM generator; questions regarding generalization across domains. **vuLZ, zgsZ, 64gQ**. | Validated results with a second LLM generator (Llama-3.2-3B) confirming the original conclusions; Added preliminary experiments applying “Math Book”-trained models to a split of CodeMMLU, with encouraging results. | Appendix E; Appendix F + Table 4. |
| **2. Methodology clarifications** | Question on whether Eval-LLM introduces new “hacking” risks; Request to clarify the trade-off between strategies C and D; **zgsZ, ChWA**. | Eval-LLM is strictly used for evaluation, so it adds no exploitation paths; Strategy D is flexible (no need to know which answer aligns with the protected feature), while C offers a more reliable signal. | Clarifications added to Section 5. |
| **3. Applicability to other settings** | Application to settings with more subtle protected features or more open-ended QA. **zgsZ, 64gQ, ChWA**. | Dealing with more subtle protected features can benefit from future work on improving automated counterfactual generation; Dealing with open-ended QA requires changing the correctness check from exact matching to LLM-as-a-judge, for example, keeping the rest unchanged. | Added discussion to Section 7. |
| **4. Presentation & Structure** | Suggestions to improve the presentation of the paper; More timely introduction of the reward model nomenclature ($RM_C$ and $RM_D$). **vuLZ, ChWA**. | Merged part of the original Figures 1 and 2; Added a dedicated illustration for the causal attribution method we use; Revised text and added additional cues for the reward model nomenclature. | Changes to Figure 1 (merged part of old Figure 2 and Figure 1); Added Figure 2; Changes to Sections 1, 3.2 and 5; Changes to some captions. |

---

Thank you once again for your time,

Authors

---

### Meta-Review · Area_Chair_Dvdm · 2026-01-11

**Summary:**

All reviewers agree the paper targets a timely failure mode: preference optimization can incentivize chain-of-thought that looks good to a reward model while not reflecting the model actual reliance on cues. The main concerns were about (1) limited scope (two controlled tasks; initially one generator LLM), (2) whether the proposed mitigation and conclusions generalize beyond stylized settings, and (3) clarity and missing definitions in the original presentation (figures, counterfactual construction, Majority@16, and RM variants). A smaller set of concerns questioned whether the evaluation setup (Eval-LLM for acknowledgment) could itself distort conclusions or introduce new “hacking” paths, and asked for controls (placebo disclaimers, robustness to prompt/decoding variation, adversarial acknowledgments).

**Reviewer Concerns:**

Presentation and missing definitions (vuLZ, ChWA): The authors reorganized the introduction and figures (merging the score distribution into Figure 1) and added a new figure explaining counterfactual construction and the causal attribution signal. They also clarified Majority@16 and added missing details about the reward models (off-the-shelf Skywork reward models and training scale).

Eval-LLM evaluation concerns (vuLZ, ChWA): They added accuracy and macro-F1 (instead of per-label F1 only) plus qualitative examples, and clarified that correctness is exact-match (no LLM judge) and Eval-LLM is evaluation-only, not part of training/selection, so it does not create a direct exploitation path. They ran a control where disclaimers are appended broadly and report it is largely ineffective, supporting the claim that gains are not purely from disclaimer semantics.

Single generator LLM (zgsZ, 64gQ): They added experiments with a second generator (Llama-3.2-3B) and report similar qualitative trends, which reduces (not eliminates) the “single model” fragility.

**Reviewer Scores:**

Reviewer vuLZ (original 4): Figures reorganized, counterfactuals explained, off-the-shelf RM details added, and evaluation metrics improved. Their remaining concern about broader domain generalization is partially addressed.

---

### Decision · Program_Chairs · 2026-01-26

Accept (Poster)